# FlowSearcher: Synthesizing Memory-Guided Agentic Workflows for Web Information Seeking

**Keyi Xiang**[1*], **Zeyu Feng**[1], **Zhuoyi Lin**[2†], **Yueming Lyu**[1], **Shi Boyuan**[1], **Yew-Soon Ong**[1,3],
**Ivor Tsang**[1,3], **Haiyan Yin**[1†]

[1]CFAR and IHPC, Agency for Science, Technology and Research (A*STAR), Singapore
[2]Institute for Infocomm Research (I[2]R), A*STAR, Singapore
[3]College of Computing and Data Science, Nanyang Technological University (NTU), Singapore

## Abstract

Web search is a cornerstone for deep research agents, enabling them to acquire and reason over knowledge beyond static corpora. Yet most existing systems rely on ReAct-style tool chains with rigid, linear workflows, hindering their ability to adapt to diverse query types and tool-use strategies. We introduce **FlowSearcher**, a novel deep search framework that formulates web information seeking as *memory-guided agentic workflow synthesis*. FlowSearcher decomposes a query into subgoals and synthesizes a tailored workflow graph for each subgoal, dynamically adapting the depth, ordering, and composition of tool use. Complementing this, a hierarchical memory consolidates past workflows into reusable structural experience, which is retrieved to guide both workflow orchestration and execution on new queries. By shifting from reactive tool calls to experience-conditioned workflow design, FlowSearcher enables flexible multi-path exploration and reuse without any supervised training or RLHF. Experiments on GAIA, BrowseComp, and GPQA show that FlowSearcher consistently matches or exceeds the performance of RLHF-trained web agents under the same model backbone. Our code is released at `github.com/XiangKeYiNTU/flowsearcher`.

## 1 Introduction

The paradigm of scaling large language models (LLMs) is shifting away from expanding static pre-training corpora toward dynamic, real-time knowledge acquisition. Central to this shift is the emergence of research agents, which couple LLMs' intrinsic reasoning capabilities with external tools and web interaction. This fusion goes beyond static retrieval, equipping models to tackle time-sensitive, knowledge-intensive tasks across domains such as science, technology, and finance (Huang et al., 2025; Xu & Peng, 2025). This trajectory is already exemplified by industrial systems such as OpenAI's Deep Research (OpenAI, 2025b;a) and Google Gemini Advanced (Comanici et al., 2025; Google, 2024), which vividly illustrate how LLMs can evolve from passive repositories into autonomous collaborators that retrieve, evaluate, and synthesize knowledge at scale.

Despite recent advances, the real bottleneck is not *model scale* but the *decision structures* that determine how agents navigate the web. Training-based systems such as WebThinker (Li et al., 2025d) and WebDancer (Wu et al., 2025a) still follow the ReAct template (Yao et al., 2023), locking agents into a think–act–observe loop. This enforces a narrow, single-threaded trajectory that collapses inherently branching research queries into linear chains, suppressing parallel exploration, backtracking, and structural revision. Plan–execute frameworks (Song et al., 2025; Zheng et al., 2025b) offer higher-level organization but remain static: once produced, the plan becomes a fixed scaffold with little room for reordering or adaptation as new evidence arrives. As a result, these architectures

---

*Work conducted while interning at Centre for Frontier AI Research (CFAR), A*STAR, Singapore.
†Corresponding authors.

remain fundamentally misaligned with the non-linear, exploratory, and continuously evolving work-flows that genuine research demands.

Another foundational challenge for long-horizon agent systems is the inability to learn across tasks (ang Gao et al., 2025). When faced with open-domain queries, most agents still operate in episode isolation (Wu et al., 2025a; Li et al., 2025a; Tao et al., 2025): tool calls are issued within short reactive chains, and whatever is learned evaporates as soon as the episode ends. This limita-tion arises from their reliance on ephemeral, episodic memory, where chains-of-thought, tool traces, and exploration paths are never consolidated into any persistent, structured knowledge. Without such consolidation, agents accumulate no reusable experience. They repeatedly reinvent the wheel, repeating ineffective actions, failing to exploit strategies that succeeded previously, and showing little improvement across similar tasks. Overcoming this bottleneck requires abandoning transient context windows in favor of a cumulative, structured memory system that can retain, organize, and reuse past workflows, supporting genuine long-horizon planning and strategic adaptation.

Building on these foundations, we introduce **FlowSearcher**, a web-based research framework that departs from the rigid, ReAct-style behavior of traditional agents by explicitly reasoning about how information-seeking procedures should be structured for a given query. Rather than committing to a single fixed tool-use trajectory, FlowSearcher represents query-conditioned workflows that allow the agent to reason about the composition, ordering, and revision of tool operations when solving complex queries. This explicit structuring enables research behaviors that are inherently non-linear, such as branching exploration, revisiting earlier decisions, and reorganizing intermediate steps as new evidence emerges. Importantly, these workflows are represented as explicit graphs, making the solution structure itself a manipulable object that can be adapted, revised, and reused across related queries, rather than an implicit byproduct of step-wise action generation.

Crucially, FlowSearcher reframes open-domain web research as **experience-driven workflow syn-thesis**, rather than sequential action prediction. Past research trajectories are not discarded after execution, but abstracted into reusable experiences that infor how future workflows should be struc-tured. By elevating structure over step-wise actions, FlowSearcher enables adaptive planning and principled revision without relying on parameter updates or supervised fine-tuning. Our results fur-ther indicate that this form of structural reuse can play a crucial complementary role to parameter-level fine-tuning in complex web information seeking, offering a path to generalization that does not depend on continual retraining. In practice, this perspective is supported by substantial empirical gains: FlowSearcher consistently outperforms strong ReAct-style baselines such as WebThinker-Base on challenging benchmarks (e.g., +11.5% on GAIA and +9.5% on BrowseComp), without task-specific parameter training.

In summary, our contributions are three-fold:

- We formulate web information seeking as a **two-level process** that separates high-level query decomposition and workflow synthesis from low-level workflow execution. This decoupling enables the system to generate query-adaptive workflows while jointly optimizing orchestration and execution for complex, multi-step search tasks.

- We reframe web search as a **workflow synthesis problem** rather than sequential tool invocation. By grounding **FlowSearcher** in a compact and expressive library of workflow building blocks, the system can flexibly compose diverse solution strategies with explicit structural control.

- We introduce a **multi-level memory** spanning node, graph, and task abstractions that consol-idates past workflows into reusable structural knowledge. A compatible retrieval and instruc-tion mechanism injects this experience into both workflow synthesis and execution, enabling memory-guided co-optimization, adaptation across tasks, and learning-free generalization.

## 2 RELATED WORK

FlowSearcher connects three lines of research: agentic information seeking, workflow planning for research agents, and long-horizon memory and experience reuse. While prior work has explored each of these directions independently, existing systems typically treat workflows as either implicit execution traces or fixed protocol templates, and memory as an auxiliary component rather than a driver of workflow design. FlowSearcher differs by unifying these strands through experience-

conditioned workflow synthesis, where accumulated execution experience directly shapes how workflows are composed and executed for new queries.

**Agentic Information Seeking Systems** Although there exist works that demonstrate different information seeking behaviors by shifting among pre-defined modes like Reason-in-Documents module introduced in Li et al. (2025c), and Problem-Solving and Report-Drafting modes introduced in Li et al. (2025d), they still adopt single-step, linear planning structures (Li et al., 2024a; Jin et al., 2025; Song et al., 2025; Zheng et al., 2025b; Wu et al., 2025a). While effective for short-horizon tasks, these approaches struggle with complex, multi-faceted queries that require branching exploration, revision, and long-range coordination across search steps. More fundamentally, their performance is often bottlenecked by the backbone LLM's ability to directly interpret and decompose diverse search queries within a single reactive trajectory. As query complexity and heterogeneity increase, this tight coupling between query understanding and step-wise action selection limits robustness, making it difficult for agents to adapt their search strategy beyond what can be implicitly encoded in the model's prompt-time reasoning.

**Workflow Planning and Optimization** Recent work explores how research agents decompose tasks and execute multi-step workflows. **ReAct-style frameworks** such as WebThinker (Li et al., 2025d) and WebWalker (Wu et al., 2025b) interleave reasoning and actions through sequential traces, but their global search strategy remains implicit, making optimization and generalization difficult. **Planning-first frameworks** (Hu et al., 2025; Tang et al., 2025b;a) impose top-down workflow structures, improving coherence but relying on manually defined roles and rules, which limits scalability in open-domain web settings (Qiu et al., 2025; Xie et al., 2025). **RL-based systems** like Web-Dancer (Wu et al., 2025a) treat information seeking as an end-to-end pipeline and optimize via sampled trajectories. Surveys such as Xu & Peng (2025), Huang et al. (2025), and Li et al. (2025b) summarize these trends. More recent systems, exemplified by AutoFlow (Li et al., 2024b), AFLOW (Zhang et al., 2025b), and MermaidFlow (Zheng et al., 2025a), move toward automated protocol or workflow synthesis, demonstrating that reusable workflow patterns can serve as transferable knowledge. However, these approaches typically operate over a predefined workflow space, relying on fixed templates, evolutionary search, or offline optimization. In contrast, FlowSearcher synthesizes workflows dynamically at inference time, conditioning both workflow structure and execution on retrieved experience and task-specific evidence.

**Agentic Memory and Experience Reuse** Another line of research equips agents with persistent memory to support long-horizon reasoning and adaptation. A-MEM (Xu et al., 2025) introduces a Zettelkasten-inspired memory system in which new experiences are stored as structured notes, linked to related past traces, and incrementally updated as new evidence arrives, enabling adaptive long-term reasoning. Mem0 (Chhikara et al., 2025) focuses on a lightweight, production-oriented memory layer that maintains persistent state across interactions and summarizes past contexts to reduce prompt length and improve personalization. G-Memory (Zhang et al., 2025a) extends memory mechanisms to multi-agent settings by organizing experiences into hierarchical graph structures at the interaction, query, and insight levels, and retrieving information at different levels of abstraction to guide coordinated agent behavior. Despite these advances, most existing memory systems rely on fixed retrieval policies or operate over a limited set of memory abstractions, which constrains their adaptability in open-domain web research. In contrast, FlowSearcher introduces a workflow-conditioned, multi-level memory hierarchy in which retrieval is jointly shaped by the task query and the evolving structure of the synthesized workflow, allowing past experience to directly inform both workflow design and execution.

## 3 METHODOLOGY

### 3.1 HIERARCHICAL AGENTIC WEB SEARCH TASK FORMULATION

In this paper, we present **FlowSearcher**, an agentic workflow framework for web search. We first reframe research query solving as a hierarchical decision-making process, where high-level query decomposition and workflow synthesis are coupled with low-level structured execution. We then introduce a memory-driven workflow planner that conditions each execution step on both workflow structure and accumulated traces. Through this design, FlowSearcher brings structural flexibility

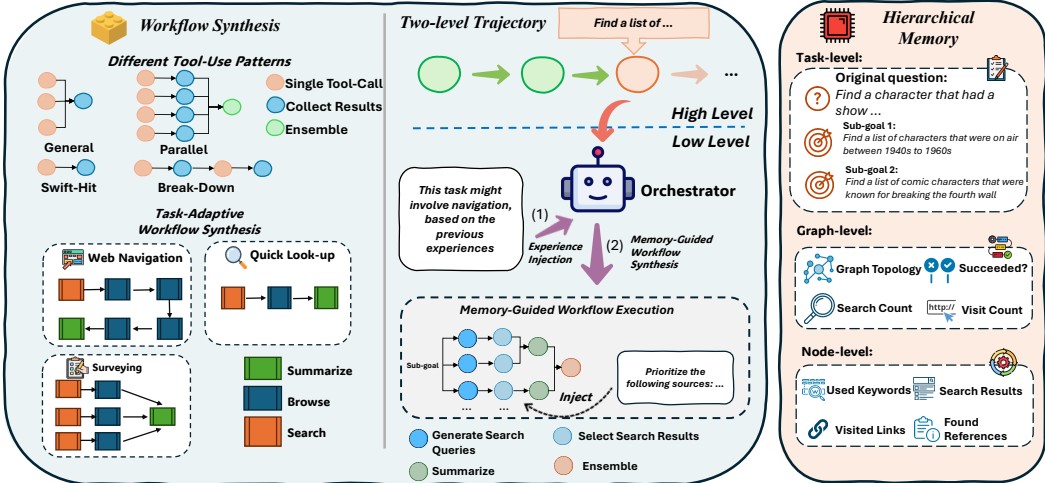

Figure 1: **An overview of the FlowSearcher framework. Left:** FlowSearcher synthesizes diverse tool-use patterns into task-adaptive workflow graphs, enabling different strategies for navigation, look-up, and surveying. **Middle:** A two-level search trajectory separates high-level orchestration from low-level execution: the orchestrator decomposes the query into sub-goals and synthesizes workflows conditioned on retrieved experience, while the executor performs memory-guided workflow execution. **Right:** A hierarchical memory records execution traces at the task, graph, and node levels, which are retrieved to guide both workflow synthesis and execution, enabling experience-driven reuse and adaptation across queries.

and layered memory grounding into agentic web search, enabling adaptive handling of complex queries and resilient reasoning over long horizons.

We formalize each research task together with its solution trajectory as $\{Q, \hat{y}, \Gamma\}$. Here, $Q$ denotes the original query, $\hat{y}$ the predicted answer, and $\Gamma = \{\mu_i, \mathcal{G}_i\}$ the trajectory consisting of decomposed sub-questions $\{\mu_i\}$ and their corresponding workflow graphs $\{\mathcal{G}_i\}$. In addition, we maintain a structured execution memory $\mathcal{M}$, which is updated after each step to record intermediate traces, and later serves as a foundation for both workflow synthesis and execution.

**High-level (decomposition $\Rightarrow$ workflow synthesis).** At the high level, FlowSearcher generates a trajectory by iteratively decomposing the query into sub-questions and synthesizing the workflow graphs. At step $i$, the agent samples the next sub-question $\mu_i$ and

$$Q \xrightarrow[\theta_\mu]{} \underbrace{\left\{ \mu_i \xrightarrow[\mathcal{M}, \theta_\mathcal{G}]{} \underbrace{\mathcal{G}_i \xrightarrow[\mathcal{M}]{} \{\alpha, o\}_v}_{\text{Lower level}} \right\}_{i=1}^{K}}_{\text{Upper level}} \xRightarrow{finalize} \hat{y}.$$

Figure 2: **FlowSearcher's hierarchical search trajectory**, involving workflow synthesis, execution, and aggregation.

then generates a workflow graph $\mathcal{G}_i$ that specifies how to address it. Formally, if the query is solved in $K$ steps, the probability of generating the overall trajectory is given by:

$$P(\Gamma \mid Q, \mathcal{M}_0) = \prod_{i=1}^{K} P(\mu_i \mid Q, \Gamma_{<i}, \mathcal{M}_{i-1}, \theta_\mu) \, P(\mathcal{G}_i \mid Q, \Gamma_{<i}, \mathcal{M}_{i-1}, \theta_\mathcal{G}, \mu_i), \qquad (1)$$

where $\theta_{\mu_i}$ and $\theta_{\mathcal{G}_i}$, are the prompts for decomposition and workflow synthesis modules. Note that at the start of each trajectory, we have $\Gamma_0, \mathcal{M}_0 = \emptyset$. As the trajectory unfolds, the memory $\mathcal{M}$ is incrementally updated, accumulating execution traces that capture both workflow structure and intermediate outcomes, providing rich contextual grounding that guides subsequent decomposition and execution. After obtaining a complete trajectory, a finalization step is performed to produce the predicted answer $\hat{y}$ for query $Q$.

**Low-level (workflow execution).** At the low level, each workflow $\mathcal{G}_i$ for sub-question $\mu_i$ is executed at the node level, where each node is followed along its dependency connections and guided

by accumulated memory traces. For a given node $v \in V(\mathcal{G}_i)$, the agent interacts with the web environment to generate an action sequence $\boldsymbol{\alpha}$ and observations $\boldsymbol{o}$. Hence, given a node with $K_v$ distinct action steps, the execution process can thus be factorized as follows:

$$P(\boldsymbol{\alpha}, \boldsymbol{o} \mid \mu_i, \mathcal{M}_{i-1}) = \prod_{t=1}^{K_v} P(\alpha_t, o_t \mid \alpha_{<t}, o_{<t}, \mu_i, \mathcal{M}_{i-1}). \tag{2}$$

Given the above formulation, the overall trajectory of FlowSearcher is illustrated in Fig. 2. Empowered by a hierarchical task structure and workflow-grounded execution, FlowSearcher departs from traditional linear tool-use agents and enables adaptive reasoning across complex query landscapes. This flexible structure not only aligns decomposition with execution, but also provides memory-grounded coherence, offering a principled path toward resilient long-horizon web search.

## 3.2 STRUCTURED COMPOSITIONAL MEMORY FOR EXPERIENCE REUSE

To enable efficient reuse of past experiences, we introduce a **Structured Compositional Memory** that organizes trajectories into a three-level hierarchy. Our design allows selective retrieval and flexible cross-level recomposition of past traces, which is crucial for adapting workflows to insightful queries and ensuring generalization beyond single-task memorization. Formally, the memory $\mathcal{M}$ is a set of task entries $\mathcal{M} = \{M_j^{task}\}$, each bundling its sub-question workflow graphs together with their node-level traces.

**Node-level.** For a node $v \in V(\mathcal{G}_i)$, we record:

$$M_v^{node} = \left(N_v, \ \boldsymbol{\alpha}^{(v)}, \ \boldsymbol{o}^{(v)}\right),$$

where $N_v$ is the node type and variant, and $\boldsymbol{\alpha}^{(v)}$ and $\boldsymbol{o}^{(v)}$ denote the action sequence and its corresponding output. This fine-grained representation enables precise replay and transfer of tool execution patterns across different sub-questions.

**Graph level.** For a workflow graph $\mathcal{G}_i$ addressing sub-question $\mu_i$, we store:

$$M_i^{graph} = \left(G_i, \ \mu_i, \ \gamma_i, \ \mathbf{n}_i, \ \{M_v^{node}\}_{v \in V(\mathcal{G}_i)}\right),$$

where $G_i$ is the textual representation of $\mathcal{G}_i$, $\gamma_i \in \{0, 1\}$ is a success indicator, and $\mathbf{n}_i \in \mathbb{N}^{|\mathcal{T}|}$ is a per-tool usage vector, with component $(n_i)_\tau$ counting how many times tool $\tau \in \mathcal{T}$ was invoked in $\mathcal{G}_i$. By recording workflow structure, performance signals, and tool statistics, graph-level memory enables targeted reuse of effective strategies while avoiding over-reliance on fragile tool chains.

**Task level.** For a query $Q_j$ we maintain:

$$M_j^{task} = \left(Q_j, \ \xi_Q, \ \{M_i^{graph}\}_{i=1}^K\right),$$

where $\xi_Q \in \{0, 1\}$ indicates whether the task is successfully solved. It encapsulates the end-to-end problem context and its outcome, allowing direct recall of solved tasks and failures alike, and providing reliable signals that guide decomposition and workflow selection in future queries.

**Memory Retrieval Mechanism.** FlowSearcher flexibly retrieves and recomposes traces from its hierarchical memory to guide agentic search with prior experience. To support adaptive reuse of trajectories, we define a unified retrieval operator $\mathcal{R}(\cdot; \zeta)$ parameterized by retrieval level $\zeta \in \{\text{graph}, \text{node}\}$. Given a new query $(Q^*, \mu^*; \zeta)$, the operator selects the top-$k$ relevant structured traces from $\mathcal{M}$:

$$\mathcal{R}(Q^*, \mu^*; \zeta) = \underset{\substack{M^{task} \in \mathcal{M}, \ Q \in M^{task}, \\ M^\zeta \subseteq M^{task}, \ \mu \in M^\zeta}}{\arg \text{top-k}} \left[\delta \frac{E(Q^*) \cdot E(Q)}{|E(Q^*)| \, |E(Q)|} + (1 - \delta) \frac{E(\mu^*) \cdot E(\mu)}{|E(\mu^*)| \, |E(\mu)|}\right]. \tag{3}$$

where $E(\cdot)$ denotes the textual embedding, and $\delta$ is a factor balancing between similarities of the original question and the sub-question. Finally, depending on the retrieval level $\zeta$, each entry is flexibly expanded within the hierarchical memory, yielding the enriched set: $\mathcal{R}(Q^*, \mu^*; \zeta) \oplus \{M_i^\zeta\}$. This flexible retrieval mechanism allows FlowSearcher to ground future reasoning not only on relevant past queries but also on useful insights extracted from targeted execution traces.

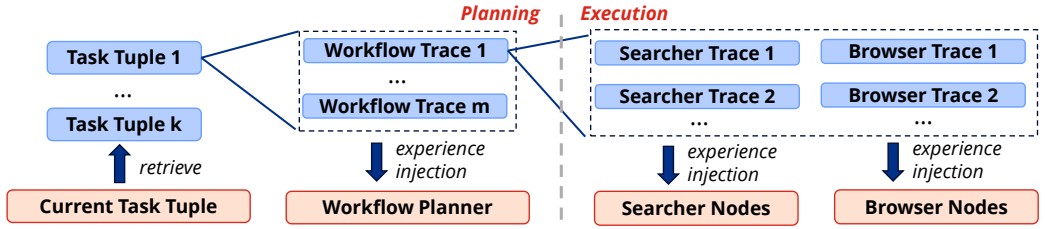

Figure 3: FlowSearcher's structured compositional memory enables the co-optimization of workflow synthesis and execution. Retrieved *task*-level tuples surface high-value workflow traces that shape the global DAG structure ("planning"), while *graph*- and *node*-level traces inject fine-grained procedural knowledge into searcher and browser nodes ("execution").

## 3.3 MEMORY-GUIDED AGENTIC WORKFLOW PLANNING

We present *memory-guided agentic workflow planning*, which aims to synthesize adaptive workflows as typed directed acyclic graphs (DAGs). In this formulation, decomposed sub-queries are handled by adaptive planning modules, while past execution traces are reused to refine and update these workflows, yielding workflows that are refined by experience at both orchestration and execution levels. Fig. 3 illustrates how FlowSearcher's hierarchical memory supports this process: workflow-level traces guide high-level DAG synthesis, whereas searcher- and browser-level traces effectively refine node behaviors and execution strategies.

At each step $i$, the orchestrator must not only generate a valid workflow, but also adapt its structure to the evolving query context. Formally, to handle sub-question $\mu_i$, it constructs a typed workflow graph as:

$$\text{orchestrator}(Q, \mu_i, \Gamma_{<i}, \mathcal{B}, \theta_{\mathcal{G}}) \xrightarrow{\mathcal{M}_{i-1}} \mathcal{G}_i\big(\mathcal{V}_{[\tau,\theta,l]}, \mathcal{E}_{[\rho]}\big),$$
$$\mathcal{V} \subseteq \mathcal{B}, \qquad \mathcal{E} \subseteq \mathcal{V} \times \mathcal{V} \text{ s.t. } (u,v) \in \rho \ \forall (u,v) \in \mathcal{E}. \tag{4}$$

Here, $\mathcal{V}_{[\tau,\theta,l]}$ denotes typed nodes parameterized by an available tool $\tau$, prompt schema $\theta$, and backbone model $l$; $\mathcal{E}_{[\rho]}$ are admissible edges constrained by the web grammar $\rho$, and $\mathcal{B}$ is the predefined set of building blocks. Crucially, the construction of $\mathcal{G}_i$ binds the query $Q$, current subquestion $\mu_i$, the previous trajectory $\Gamma_{<i}$, and memory $\mathcal{M}_{i-1}$. This makes each workflow not just a static composition, but an *experience-driven program* that evolves with context.

**Experience-Guided Workflow Orchestration.** Retrieved graph-level traces inject prior experience into orchestration, exposing both successful and failed strategies, tool-usage statistics, and graph topologies. Formally,

$$\tilde{M}_{\mathcal{G}} = R_{\mathcal{G}}(Q^*, \mu^*), \qquad \xi_{\mathcal{G}} = I_{\mathcal{G}}(\tilde{M}_{\mathcal{G}}, Q^*, \mu^*), \quad \mathcal{G}^* = \text{orchestrator}(\theta_{\mathcal{G}} \oplus \xi_{\mathcal{G}}). \tag{5}$$

Here $\tilde{M}_{\mathcal{G}} = \{\mathcal{G}, \gamma, \mathbf{n}^k\}_{1..K}$ denotes the retrieved graph-level traces, each containing the workflow structure $\mathcal{G}$, its success indicator $\gamma$, and tool-usage statistics $\mathbf{n}^k$. These traces are expanded into full execution records and distilled by the orchestration instructor $I_{\mathcal{G}}$ into concise insights $\xi_{\mathcal{G}}$, which are then injected into the orchestration prompt $\theta_{\mathcal{G}}$. As a result, the workflow $\mathcal{G}^*$ is not a static composition, but one shaped by prior evidence: by contrasting successful and unsuccessful workflows, FlowSearcher uncovers structural patterns that guide effective design choices, while tool-usage statistics across topologies reveal how efficiency scales with structure. These insights ground orchestration in empirical evidence, making it adaptive, resource-aware, and systematically refined by past executions.

**Experience-Guided Workflow Execution.** We formalize the node-level memory-enhanced execution process as equation 6:

$$\tilde{M}_v = \mathcal{R}_v(Q^*, \mu^*), \quad \xi_v = I_v(\tilde{M}_v, \mathcal{G}^*, Q^*, \mu^*), \quad (\boldsymbol{\alpha}^*, \boldsymbol{o}^*) = \text{execute}(\theta_v \oplus \xi_v, \tau_v). \tag{6}$$

The retrieved traces, $\tilde{M}_v = \{N, (\boldsymbol{\alpha}, \boldsymbol{o})\}_{1..K}$, expand into execution logs that capture node configuration $N$, action sequences $\boldsymbol{\alpha}$, and their outcomes $\boldsymbol{o}$. Distilled by the node instructor $I_v$, these traces

Table 1: Performance comparisons on three benchmarks. We report Pass@1 metric on all tasks. The best results are highlighted in **bold** and the first runner-ups are underlined. Results from OpenAI's Deep Research are presented in gray for reference.

| Backbone | Framework | GAIA | | | | GPQA-Diamond | | | | BrowseComp | | |
|---|---|---|---|---|---|---|---|---|---|---|---|---|
| | | Level 1 | Level 2 | Level 3 | Avg. | Phy. | Chem. | Bio. | Avg. | Art | His. | Avg. |
| *No Agency* | | | | | | | | | | | | |
| Qwen-2.5-32B | Base | 20.5 | 9.6 | 8.3 | 13.1 | 52.3 | 30.1 | 68.4 | 43.4 | 0.0 | 0.0 | 0.0 |
| | RAG | 20.3 | 11.8 | 6.3 | 13.0 | 64.0 | 41.9 | 57.9 | 53.0 | 0.0 | 0.0 | 0.0 |
| Qwen-2.5-72B | Base | 20.5 | 13.5 | 6.0 | 13.4 | 58.1 | 39.8 | 57.9 | 49.5 | 0.0 | 0.0 | 0.0 |
| GPT-4o | Base | 23.1 | 15.4 | 8.3 | 15.6 | 62.8 | 46.2 | 68.4 | 55.6 | 0.8 | 0.8 | 0.8 |
| QwQ-32B | Base | 30.8 | 15.6 | 6.7 | 17.7 | 84.8 | 44.1 | 68.4 | 64.1 | 0.0 | 0.0 | 0.0 |
| | RAG | 33.3 | 25.0 | 0.0 | 19.4 | 84.9 | 45.2 | 73.7 | 65.2 | 0.0 | 0.0 | 0.0 |
| DeepSeek-R1-671B | Base | 43.6 | 26.9 | 8.3 | 31.1 | **90.7** | **57.0** | **84.2** | **74.2** | 0.0 | 0.0 | 0.0 |
| *Close-Sourced Agentic Frameworks* | | | | | | | | | | | | |
| OpenAI DR | | 74.3 | 69.1 | 47.6 | 67.4 | - | - | - | - | - | - | 51.5 |
| *ReAct Agentic Frameworks* | | | | | | | | | | | | |
| Qwen-2.5-32B | Vanilla ReAct | 46.1 | 26.9 | 0.0 | 31.0 | 64.0 | 41.9 | 57.9 | 53.0 | 0.0 | 0.0 | 0.0 |
| | WebDancer | 46.1 | 44.2 | 8.3 | 40.7 | - | - | - | - | - | - | - |
| QwQ-32B | Vanilla ReAct | 48.7 | 34.6 | 16.6 | 37.8 | 76.7 | 46.2 | 68.4 | 61.6 | 0.8 | 0.0 | 0.4 |
| | Search-o1 | 61.5 | 50.0 | **25.0** | 51.5 | 77.9 | 47.3 | 78.9 | 63.6 | 1.6 | 2.4 | 1.9 |
| | WebThinker-Base | 53.8 | 44.2 | 16.7 | 44.7 | 87.2 | 51.6 | 68.4 | 68.7 | 2.4 | 2.4 | 2.3 |
| | WebThinker-RL | 56.4 | 50.0 | 16.7 | 48.5 | 90.7 | 50.5 | 78.9 | 70.7 | 2.4 | 3.1 | 2.7 |
| | WebDancer | 61.5 | 50.0 | 25.0 | 51.5 | - | - | - | - | - | - | 3.8 |
| *Ours* | | | | | | | | | | | | |
| Qwen-2.5-32B | **FlowSearcher** | 61.5 | 46.2 | 16.7 | 48.5 | 72.0 | 47.3 | 68.4 | 60.1 | 5.5 | 5.6 | 5.6 |
| Qwen-3-32B | **FlowSearcher** | **69.2** | 53.8 | 16.7 | 55.3 | 87.2 | 48.4 | 78.9 | 68.2 | 8.7 | 8.0 | 8.1 |
| QwQ-32B | **FlowSearcher** | 66.7 | **57.7** | 16.7 | 56.3 | 90.7 | 51.6 | 78.9 | 71.2 | 7.9 | 7.2 | 8.0 |
| GPT-4o-mini | **FlowSearcher** | 66.7 | 53.8 | **25.0** | 55.3 | 81.4 | 49.5 | 73.7 | 65.7 | 11.0 | 12.0 | **11.8** |

yield execution insights $\xi_v$, which are injected into the node prompt $\theta_v$ to guide the next action sequence $(\boldsymbol{\alpha}^*, \boldsymbol{o}^*)$.

Beyond replay, these traces enable *node-type specialization*, refining execution strategies for roles such as retrieval, parsing, or tool invocation. They also support *cross-query transfer*, allowing new tasks to inherit behaviors from structurally similar nodes. Crucially, the workflow graph $\mathcal{G}^*$ provides the scaffold for structure, while node-level memory drives local behavioral refinement. This division localizes and mitigates errors at the node level, while improving robustness over long-horizon workflow executions.

Together with orchestration, this node-level adaptation achieves the *co-optimization of workflow planning and execution*, ensuring workflows evolve holistically with both structural and behavioral guidance. Beyond conventional rigid workflows that are statically designed and replayed, FlowSearcher enables adaptive strategies that scale to diverse tasks and promotes transfer across queries by grounding decisions in accumulated experience. To the best of our knowledge, FlowSearcher is the first framework to realize such experience-driven agentic workflow planning.

# 4 EXPERIMENTS

## 4.1 EXPERIMENT SETUP

**Tasks and Benchmarks.** We evaluate **FlowSearcher** on three challenging benchmarks: **GAIA:**(Mialon et al., 2023) A benchmark testing AI models' ability as general assistants with three levels of difficulty, we used 103 text-only questions to evaluate our system's complex information retrieval and reasoning abilities; **BrowseComp:**(Wei et al., 2025) A benchmark for browsing agents consisting of "hard to solve yet easy to verify" questions across topics like Art, History, etc.; **GPQA-Diamond:**(Rein et al., 2023) A graduate-level Google-proof benchmarks of multi-choice questions across three domains: Physics, Biology, and Chemistry. We chose the Diamond subset as our test set, which consists of questions experts can answer correctly but non-experts rarely.

**Baselines.** We compare **FlowSearcher** with three types of baselines: (1) Vanilla LLMs with no agency: under this category, we tested two variants: Base LLMs with no access to search tools and LLMs incorporated with standard RAG which retrieved top-10 relevant documents from search engines as references before generating answers. (2) Close-sourced proprietary framework: We

chose OpenAI Deep Research(OpenAI, 2025b) as an example of commercial solution. Notably, we exclude it from our quantitative comparisons because its full methodology, training setup, and evaluation pipeline are not publicly available, making results non-reproducible. (3) Existing agentic frameworks (particularly ReAct-style): including Vanilla ReAct, WebThinker(Li et al., 2025d), WebDancer(Wu et al., 2025a), and Search-o1(Li et al., 2025c).

**Implementation Details.** We utilized a variety of LLM backbones, including both open-sourced models (Qwen3-32B, Qwen2.5-32B, QwQ-32B(Yang et al., 2025a;b)) and closed-sourced models (GPT-4o-mini(OpenAI, 2024)). We used SerpAPI* to enable web search and the Jina Reader API† for browsing capabilities. For each query initiated by the agents, we retrieved the top 10 results from the respective search tool. We provide details of prompts and data schemas as per Appendix C.

## 4.2 BENCHMARK EVALUATION RESULTS

We present benchmark evaluation results in Table 1.

ReAct-style agents do provide noticeable gains. For instance, vanilla ReAct boosts Qwen2.5–32B's GAIA score by **+25.8** over standard RAG. However, their rigid step-wise structure limits further improvement. Even with reinforcement learning, progress quickly plateaus: WebThinker-RL improves over WebThinker-Base by only **+3.8** on GAIA, **+2.0** on GPQA, and a negligible **+0.2** on BrowseComp, despite the heavy cost of dataset construction and training. These results reveal a core constraint: *fine-tuning alone cannot overcome the structural bottlenecks of linear ReAct-style search*.

In contrast, **FlowSearcher** avoids these limitations through dynamic, memory-guided workflow synthesis. Without any supervised fine-tuning, it consistently outperforms comparably scaled agentic baselines. With a QwQ–32B backbone, FlowSearcher surpasses WebDancer by **+4.8** on GAIA and **+4.2** on BrowseComp. Its advantage becomes even clearer on BrowseComp, which stresses open-domain browsing and long-horizon reasoning: FlowSearcher achieves a further **+8.0%** improvement using a GPT–4o-mini backbone. On GPQA-Diamond, it reaches performance competitive with advanced reasoning models such as DeepSeek-R1–671B.

These results illuminate two key insights: **(i) FlowSearcher's structural flexibility enables it to adapt reasoning procedures to large, noisy, and unpredictable information spaces**, a capability that rigid ReAct-style workflows struggle to match. **(ii) FlowSearcher improves purely through experience-driven workflow planning, without relying on any fine-tuning pipeline**. This independence makes the system practical, generalizable, and naturally self-refining: it continually strengthens its workflows directly from execution history. In open-domain knowledge environments, FlowSearcher simply has more strategic leverage to navigate diverse and unreliable sources of information.

## 4.3 BLOCK USAGE

In this section, we evaluate the functional contributions of the pre-defined modules in **FlowSearcher**. We begin by analyzing block usage on the GAIA benchmark with the GPT-4o-mini backbone, verifying that the workflow orchestrator assigns each block to tasks aligned with its intended role.

The block usage across GAIA's three levels is shown in Fig. 4. Our key observations are:

**Searcher block usage patterns are largely consistent across levels.** Among the variants, the first-hit searcher dominates at all levels, reflecting that most sub-steps involve quick look-ups. However, parallel searchers appear more frequently in Levels 2 and 3, capturing the added complexity of harder problems.

**Browser block usage varies significantly.** At Level 1, the first-hit browser was dominant, often paired with first-hit searchers for simple fact-checking. In contrast, usage of the in-depth browser rose sharply at higher levels, becoming the most frequently used at Level 3. This indicates that more complex tasks at Levels 2 and 3 required deeper web navigation and interaction with webpages.

---

*https://serpapi.com/
†https://jina.ai/reader/

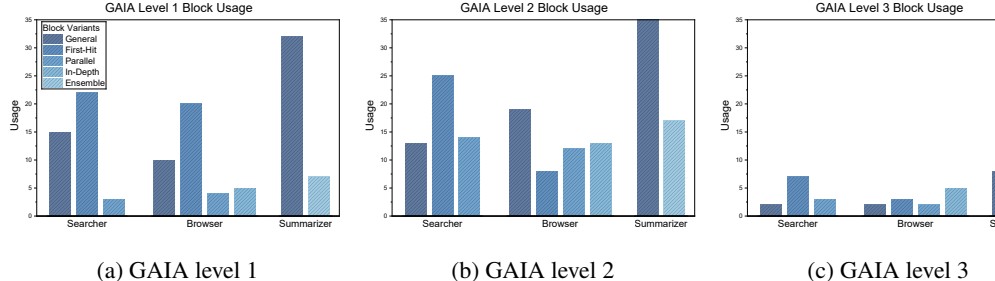

|  (a) GAIA level 1 | (b) GAIA level 2 | (c) GAIA level 3 |

Figure 4: Usage of each block variant on GAIA evaluation tests with GPT-4o-mini backbone. The distribution highlights how the orchestrator assigns blocks in alignment with their intended roles, with deeper browser variants increasingly favored at higher task levels.

**Summarizer block usage is the most stable.** Between the two summarizer variants, the general summarizer was consistently more common than the ensemble summarizer across all levels. Nonetheless, a slight increase in ensemble summarizer usage was observed at Levels 2 and 3 compared to Level 1.

These studies reveal that **FlowSearcher** adapts its strategies to the blocks at hand, shifting from quick look-ups to deeper browsing when searchers are limited, or leaning on summarization when browsing capacity is reduced. Crucially, even with a smaller toolset, the system reorganizes workflows to sustain performance, underscoring its adaptability and robustness under constrained searching conditions.

## 4.4 ABLATION STUDY

In this section, we conduct ablation studies that probe the internal mechanics of FlowSearcher by independently scaling its two core modules: (a) **block library**, which governs the expressiveness of workflow synthesis; (b) **hierarchical memory**, which governs experience-driven refinement.

### 4.4.1 IMPACT OF SCALING THE BUILDING BLOCK LIBRARY

For analyzing the effects of scaling blocks, we conducted three groups of controlled experiments on GAIA with GPT-4o-mini backbone, where the set of available blocks was configured following three settings:

**First-Hit Only.** In this condition, the searcher and browser modules are restricted to their first-hit variants exclusively. Accordingly, the system is permitted to perform only a single search query at a time, and the browsing process terminates immediately once the first relevant piece of information is retrieved.

**First-Hit + General.** In this condition, the range of available modules is expanded to include both the general and first-hit variants of the searcher and browser. The system may therefore issue up to five search queries and aggregate their results, and it may browse up to ten distinct webpages while extracting relevant information. However, the system can't conduct in-depth browsing, which means they cannot click links on pages and perform web navigation tasks.

**No Limitations.** In this condition, the orchestrator operates without any restrictions on module selection. All block types are available, and the system may employ them without predefined limits. This represents the default, unconstrained configuration, supporting the broadest possible range of retrieval strategies.

The results (Table 2) show a clear progression. The *first-hit only* condition struggled due to its rigid constraints. Adding general blocks yielded a moderate gain of **+7.8%**, while lifting all restrictions produced a substantial **+20.3%** improvement. These numbers underscore a simple truth: limiting core building blocks narrows workflow flexibility, whereas expanding the available toolkit unlocks richer, more effective execution patterns, mirroring real-life web search, where diverse strategies are often required, much like how humans adapt their browsing behavior to the task at hand. This progression also indicates that workflow expressiveness, rather than raw tool count, is a key driver

Table 2: Performance on different sets of block options (with increasing system-level flexibility). Results show that broader block availability consistently enables more diverse strategies and delivers higher performance.

| Block Option | GAIA | | | |
| --- | --- | --- | --- | --- |
| | Level 1 | Level 2 | Level 3 | Avg. |
| First-Hit only | 35.9 | 26.9 | 0.0 | 27.2 |
| First-Hit + General | 41.0 | 36.5 | 8.3 | 35.0 |
| No limitations | **66.7** | **53.8** | **25.0** | **55.3** |

of performance, as richer block combinations allow the orchestrator to better align search strategies with query structure.

### 4.4.2 IMPACT OF MEMORY COMPOSITION

In this section, we conducted four groups of controlled experiments to study the impact of utilizing different memory composition: **(i) No Memory:** FlowSearcher synthesizes and executes workflows with no memory retrieval and experience injection in this settings; **(ii) Full Memory:** FlowSearcher synthesizes and executes workflows while recording and utilizing all past traces; **(iii) Only Successful Memory:** Only successful episodes are recorded and retrieved; **(iv) Only Unsuccessful Memory:** Only unsuccessful episodes are recorded and retrieved.

Table 3: Cumulative number of successful tasks under different memory compositions.

| Task Window | Full Mem. | No Mem. | Succ.- Only | Unsucc.- Only |
| --- | --- | --- | --- | --- |
| **1-20** | 5 | 7 | 6 | 5 |
| **1-40** | 16 (+11) | 18 (+11) | 20 (+14) | 13 (+8) |
| **1-60** | 26 (+10) | 24 (+6) | 30 (+10) | 24 (+11) |
| **1-80** | 40 (+14) | 33 (+9) | 42 (+12) | 36 (+12) |
| **1-103** | 57 (+17) | 42 (+9) | 53 (+11) | 48 (+12) |

We shuffled GAIA's 103 tasks in order to observe the unbiased trend shown in Table 3. We inferred from the results that: (a) **Successful-only memory yields the fastest early-stage gains** because it reinforces high-quality positive patterns without noise; (b) **Full memory eventually overtakes all others**, as combining successful and unsuccessful traces enables stronger long-term correction and generalization; (c) **No-memory and unsuccessful-only strategies improve far more slowly**, highlighting the importance of structured experience reuse for continual self-improvement. Overall, these trends reveal a clear trade-off between exploitation and correction: prioritizing successful traces accelerates early learning by stabilizing effective patterns, while incorporating unsuccessful experiences becomes critical over time to expose failure modes, refine workflow structure, and support robust long-horizon generalization.

## 5 CONCLUSION

In this work, we introduced **FlowSearcher**, a framework that redefines web information seeking through experience-driven agentic workflows. Rather than relying on reactive tool-use, FlowSearcher constructs and optimizes full workflow graphs, supported by a structured memory that retrieves and adapts past trajectories across tasks. These reusable traces directly inform both workflow orchestration and execution, enabling FlowSearcher to achieve consistent and sizable gains over strong baselines on three challenging benchmarks. Beyond empirical results, our findings highlight a broader insight: **memory-driven workflow design can unlock improvements on par with, and in some cases exceeding, those achieved through conventional fine-tuning**. More broadly, our results suggest that elevating workflow structure and experience reuse to first-class design principles offers a scalable alternative to purely parametric approaches for building robust, long-horizon research agents. Looking forward, we aim to enrich FlowSearcher's memory with finer-grained patterns and distilled abstractions, and to extend its workflow representations with more expressive structures, moving toward agents that are increasingly adaptive, transferable, and self-improving.

## ACKNOWLEDGMENTS

This research is supported by the National Research Foundation, Singapore under its AI Singapore Programme (AISG Award No: AISG-NMLP-2024-003), and the National Research Foundation, Singapore and Infocomm Media Development Authority under its Trust Tech Funding Initiative, Career Development Fund (CDF) of the Agency for Science, Technology and Research (A*STAR) (No: C243512014). Any opinions, findings and conclusions or recommendations expressed in this material are those of the author(s) and do not reflect the views of the National Research Foundation, Singapore, the Agency for Science, Technology and Research, or the Infocomm Media Development Authority.

## REPRODUCIBILITY STATEMENT

To ensure the reproducibility and transparency of our research findings, all evaluations were conducted using publicly available benchmarks: GAIA (General AI Assistant benchmark)[‡], GPQA-Diamond (Graduate-level Google-Proof Q&A benchmark)[§], and BrowseComp (Web browsing and comprehension benchmark)[¶]. These datasets are openly accessible to the research community, enabling replication of our experimental conditions. Environment setting scripts, benchmark reproducing scripts (testing and evaluation) are provided as per in the Supplementary Materials. Comprehensive implementation details are provided in the Appendix. This documentation, combined with the standardized nature of the public benchmarks, ensures that our work can be independently reproduced and validated by the research community.

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

# A DEFINITIONS AND NOTATIONS

The notations and definitions are presented in Table 4.

Table 4: Notations and definitions used in the FlowSearcher methodology.

| Notation | Definition |
|---|---|
| $Q$ | The original query (main research question). |
| $\hat{y}$ | The predicted answer aggregated from workflow executions. |
| $\mu_i$ | Sub-question at step $i$, decomposed from $Q$. |
| $\Gamma = \{\mu_i, \mathcal{G}_i\}$ | Solution trajectory consisting of sub-questions and their workflow graphs. |
| $\mathcal{G}_i$ | Workflow graph composed of building blocks (e.g., search, browse, summarize). |
| $v = (\boldsymbol{\alpha}, \boldsymbol{o})$ | Node representation: action sequence $\boldsymbol{\alpha}$ and corresponding outputs $\boldsymbol{o}$. |
| $\mathcal{M}$ | Structured hierarchical memory storing past task, graph, and node traces. |
| $\theta_\mu$ | Prompt for sub-question decomposition. |
| $\theta_\mathcal{G}$ | Prompt for workflow synthesis. |
| $P(\Gamma \mid Q, \mathcal{M}_0)$ | Probability of generating a trajectory given query $Q$ and initial memory $\mathcal{M}_0$. |
| $P(\boldsymbol{\alpha}, \boldsymbol{o} \mid \mu_i, \mathcal{M}_{i-1})$ | Probability of node execution (action-output sequence) conditioned on sub-question and memory. |

# B INFERENCE PROCESS

This section we are going to walk through a typical pipeline of solving a task by FlowSearcher.

## B.1 HIGH LEVEL: NEXT SUB-QUESTION GENERATION AND MEMORY-GUIDED WORKFLOW SYNTHESIS

**Sub-question generation and experience retrieval.** The inference procedure is organized in a stepwise manner. At the beginning of each step, a sub-question is generated based on the aggregated observations and the original query. To guide the orchestration, three entries from the execution log are retrieved from memory. These logs are processed and passed to the instructor module, which distills them into at most three concise and transferable experiences on workflow orchestration. The resulting experiences are then incorporated into the orchestrator prompt.

**Workflow graph orchestration and validation.** Once the workflow graph, represented in YAML, is orchestrated, the system's graph validation module is invoked to verify whether the generated graph is valid (see Section C.2). If validation fails, the orchestrator is prompted to regenerate the graph using the provided error message.

**Conversion to executable code.** After a valid workflow is obtained, the converter module translates the YAML graph specification into executable Python files.

**Workspace creation.** Finally, a workspace folder is created, into which all relevant files are stored, including the pre-execution metadata (original question, current sub-question, aggregated observations, and tool-call details), the YAML configuration file, and the generated executables.

A high-level process demonstration is presented in Algorithm 1.

---

**Algorithm 1** FlowSearcher High-Level Inference Process

---

**Require:** Original Query $Q_{orig}$
**Ensure:** Final Answer $A_{final}$
1: Initialize Aggregated Observations $O \leftarrow \emptyset$
2: $final\_answer\_found \leftarrow$ false
3: **while not** $final\_answer\_found$ **do**
    *— Sub-question Generation and Memory-Guided Workflow Synthesis —*
4:      Generate sub-question $Q_{sub}$ based on $Q_{orig}$ and $O$.
5:      Retrieve execution logs from memory and distill into experiences $E$.
    *— Workflow Graph Orchestration and Validation —*
6:      **repeat**
7:        Orchestrate workflow graph $G$ (YAML) using $Q_{sub}$ and $E$.
8:        $is\_valid, error\_msg \leftarrow$ ValidateGraph($G$).
9:        **if not** $is\_valid$ **then**
10:          Regenerate $G$ with feedback from $error\_msg$.
11:        **end if**
12:      **until** $is\_valid$
    *— Conversion and Workspace Setup —*
13:     Convert valid graph $G$ into executable Python files $F_{exec}$.
14:     Create workspace and store metadata, $G$, and $F_{exec}$.

15:     Call EXECUTEWORKFLOW($Q_{sub}, F_{exec}, O$)

    *— Final Verification —*
16:     Determine if $Q_{orig}$ can be answered with the updated observations $O$.
17:     **if** $Q_{orig}$ is answerable **then**
18:       $final\_answer\_found \leftarrow$ true
19:     **end if**         ▷ If not, the loop continues to generate the next sub-question.
20: **end while**
    *— Execution and Result Aggregation —*
21: Invoke the **Finalizer Block**.
22: Identify the most relevant references $R_{rel}$ within $O$.
23: Verify factual consistency between $R_{rel}$ and the summary in $O$.
24: Synthesize the final answer $A_{final}$ by reasoning over the verified summary.
25: **return** $A_{final}$

---

### B.2 Low Level: Memory-Guided Workflow Execution

**Execution of Searcher Blocks.** Searcher blocks are responsible for generating search keywords, performing searches, and selecting relevant search results. URLs from the selected results are passed directly to the browser blocks for further processing.

**Execution of Browser Blocks.** Browser blocks navigate the webpages provided by the searcher blocks and extract information from them. In the case of in-depth browsing, additional links on the page are also collected. All extracted content is organized as a list of references, each containing the original information and its source URL.

**Execution of Summarizer Blocks.** Summarizer blocks select the most relevant references and produce a concise summary that addresses the current sub-question and contributes to the overall answer. The updated summary is then stored for subsequent use.

**Verification Process**

1. **Sub-question Verification:** Check if the current sub-question has been successfully addressed.
   - If verification fails, repeat the workflow until conditions are met.

2. **Final Verification:** If the sub-question is verified, determine whether the original question can now be answered:
   - If yes, activate the aggregation module to produce the final answer.
   - If no, generate the next sub-question and continue the high-level control flow.

We present the low-level process as shown in Algorithm 2:

---

**Algorithm 2** FlowSearcher Low-Level Workflow Execution

---
1: **procedure** EXECUTEWORKFLOW($Q_{sub}, F_{exec}, O$)
2:    **repeat**
   *— Block Execution —*
3:       **Searcher Blocks:** Generate keywords, perform search, and select results (URLs).
4:       **Browser Blocks:** Navigate URLs, extract content, and organize as references.
5:       **Summarizer Blocks:** Select relevant references and generate a summary for $Q_{sub}$.
6:       Store the new summary and references in the aggregated observations $O$.
   *— Sub-question Verification —*
7:       Check if $Q_{sub}$ has been successfully addressed by the generated summary.
8:    **until** $Q_{sub}$ is addressed
9: **end procedure**

---

### B.3 Execution and Result Aggregation

When the accumulated summary and references pass the final verification stage, **the finalizer block** is invoked to derive a concise answer to the original query. The finalizer operates in two steps. First, it identifies the subset of references most relevant to the query and verifies the factual consistency between these references and the constructed summary. Subsequently, it reasons over the verified portion of the summary and synthesizes a concise, accurate answer.

## C Implementation Details

**FlowSearcher** is implemented through Langgraph[‖]. In this section, we present the implementation details of **FlowSearcher**.

### C.1 Building Blocks

In **FlowSearcher**, building blocks are categorized into these types:

---

[‖]https://www.langchain.com/langgraph

1. **Searcher:** Responsible for retrieving relevant information from external sources, such as databases, search engines, or knowledge bases, based on the current query or sub-question. It provides the raw material for further reasoning and analysis.

2. **Browser:** Navigates through the retrieved resources to extract structured and unstructured information. The browser interprets web pages, documents, or other content, and transforms them into a form usable by downstream modules.

3. **Summarizer:** Condenses the collected information into concise, coherent summaries. It filters out irrelevant details, highlights key points, and prepares the content for verification and higher-level reasoning.

4. **Verifier:** Checks the factuality and consistency of the summarized content against the original sources or cross-references. It ensures that the information used for reasoning is accurate and trustworthy.

5. **Finalizer:** Integrates verified information to produce a coherent and concise answer to the main query. The finalizer synthesizes evidence from multiple sources and ensures that the resulting answer is accurate, complete, and well-structured.

6. **Thinker:** Performs high-level reasoning and problem-solving. It can generate sub-questions, plan multi-step workflows, and determine which tools or blocks should be invoked to solve complex tasks.

Now we are presenting the two most important functional block types: Searcher and Browser.

### C.1.1 SEARCHER-TYPE BLOCKS

The searcher blocks receive the original question (`OverallState.messages[0].content`) and the current_summary (`OverallState.current_summary`) as input. Upon completion of their execution, all fields within `SearcherState` are updated to reflect the results of the search process. The state structure is demonstrated in C.4.

**General Searcher** first generates up to 5 search queries, performs a search and collects search results for each query. The prompts for generating search queries and collecting search results for each query are as follows:

---

**Search Query Generation Prompt**

You are a query writer agent that operates in a workflow that solves a question step by step.

You are given:
- The main question
- The sub-goal of the current step
- Some used keywords or phrases used in the previous searches
- A summary containing current found information

**Your tasks:**
- The current summary (if provided) fails to reach the sub-goal
- Output some keywords or phrases that have the potential to find other useful information outside of the current summary and related to the sub-goal
- Don't output more than {query_count} keywords or phrases

**Extra notes:**
- If no used keywords and summary provided, that means you need to think about the first keywords to search
- The current date is **current_date**, be careful when it's necessary to specify time in the search keywords
 experiences

 —

 Main question:
{original_question}
 Sub-goal:

---

{sub_question}
Used search queries:

{used_search_keywords_and_phrases}
Current summary:

{current_summary}

---

**Search Result Selection Prompt**

You are a search result selection agent that operates in a workflow that solves a question step by step.

You are given:
- The main question
- The sub-goal of the current step
- Search results obtained by searching for {query}

**Your tasks:**
- Select relevant search results and only output their URLs and snippets
- If no relevant search results provided, output an empty list
- The current date is **{current_date}**, be careful when the question requires updated information
{experiences}

—

Main question:

{original_question}
Sub-goal:

{sub_question}
Search results:

{search_results}

---

**First-Hit Searcher** only generates one search query and performs one search, the search result selection process is the same as General Searcher. The query generation prompt is as follows:

---

**Goal Break-Down Prompt**

You are a query writer agent that operates in a workflow that solves a question step by step.
You are given:

- The main question
- The sub-goal of the current step
- Some used keywords or phrases used in the previous searches
- A summary containing current found information
**Your tasks:**

- The current summary (if provided) fails to answer the sub-goal
- Output one search query that has the potential to find other useful information outside of the current summary and related to the sub-goal
**Extra notes:**

- If no used keywords and summary provided, that means you need to think about the first keyword to search
- The current date is **{current_date}**, be careful when it's necessary to specify time in the search keyword

---

{experiences}
—

Main question:
{original_question}
Sub-goal:
{sub_question}
Used search queries:
{used_search_keywords_and_phrases}
Current summary:
{current_summary}

**Parallel Searcher.** Specifically, given a sub-goal, the block first decomposes it into a structured list of finer-grained goals that can each be independently addressed. For every goal, the block automatically generates one or more search queries tailored to the goal's intent and retrieves the corresponding candidate results. To ensure consistency and reliability, all retrieved outputs are subsequently processed through a unified result selection procedure that ranks, filters, and consolidates the candidate results into a coherent evidence set. This design enables the block to operate as a self-contained unit that bridges abstract sub-goals with concrete, high-quality information.

### Break-down Goal Prompt

You are a helper agent breaking down a goal into a list of ready-to-search sub-goals that operates in a workflow that solves a question step by step. The workflow is solving the question using a search engine.
 You are given:

- The main question
- The sub-goal of the current step
 **Your tasks:**

- Break down the current sub-goal into a list of ready-to-search sub-goals
- If the sub-goal is already specific enough to conduct a search on it, just output the sub-goal as a single item in the list
- The current date is **{current_date}**, be careful when the question requires updated information
 —

 Main question:

{question}
 Sub-goal:

{sub_goal}
 —

 Example:

Main question: "Help me find a character who constantly breaks the fouth wall and has a backstory of being saved by an ascetic"
 Sub-goal: "Find which characters from this list have a backstory of being saved by an ascetic: A, B, C, D, E"
 List of ready-to-search sub-goals:

"Find A's backstory and determine if A is saved by an ascetic",
"Find B's backstory and determine if B is saved by an ascetic",
"Find C's backstory and determine if C is saved by an ascetic",
"Find D's backstory and determine if D is saved by an ascetic",
"Find E's backstory and determine if E is saved by an ascetic"

### C.1.2 BROWSER-TYPE BLOCKS

The browser blocks receive search results from the orchestrated searcher blocks, visit the URLs, and extract relevant information. Webpage content is split into chunks, and at each step, browser blocks must decide whether to continue visiting the next chunk.

**General Browser.** The general browser block selects up to five URLs and sequentially extracts information from each page it visits.

---

**URL Selection Prompt**

You are an agent that selects next relevant URLs to browse when solving a question step by step.
 You are given:

- The main question
- The sub-goal of the current step
- A list of URLs and their snippets
 **Your tasks:**

- Check through the list of URLs and their snippets and determine what kind of information is being provided relevant to the sub-goal
- Select the URLs that have the potential to provide useful information relevant to the sub-goal, you can select them all if you think they are all relevant
- The current date is **{current_date}**, be careful when the question requires updated information
 **Extra notes:**

- **CRITICAL: Pay close attention to specific requirements in the question** (e.g., "official script", "official website", "primary source", "government data", etc.)
- **Prioritize URLs that match the specific source requirements mentioned in the question**
- If the question asks for "official" sources, prioritize URLs from official organizations, government sites, or primary sources over fan sites, transcripts, or secondary sources
 **Source Priority Guidelines:**

- Official/Primary sources: Government sites (.gov), official organization websites, original publishers, etc.
- Secondary sources: News sites, academic sites, established databases
- Tertiary sources: Fan sites, transcripts, wikis, forums (use only if no better sources available)
 {experiences} —

 Main question:

{original_question} Sub-goal:

{sub_question} List of URLs and their snippets:

{list_of_urls_and_snippets}

---

**Information Extraction Prompt**

You are an information extractor agent that operates in a workflow that solves a question step by step.
 You are given:

- The main question
- The sub-goal of the current step
- Part of the content of a webpage, you will be given the number of parts and the index of the current part
 **Your tasks:**

- Extract ONLY information that is directly relevant to answering the sub-goal or the main question
- Be selective and focused - avoid extracting tangential information like version histories, contributor lists, or general background unless specifically needed

---

- The current date is **{current_date}**, be careful when the question requires updated information
 **What NOT to extract:**
- Version histories or release notes unless the question specifically asks about versions
- Contributor lists or acknowledgments unless the question asks about contributors
- General background information that doesn't directly relate to the question
- Marketing content, testimonials, or promotional material
- Navigation elements, headers, footers, or UI text
- Repeated information that has already been captured

—
Main question:
{original_question} Sub-goal:
{sub_question} Webpage content:
{webpage_content}

**First-Hit Browser.** The first-hit browser block selects the single most reliable URL and stops immediately after retrieving the relevant information.

---

**First-hit Information Extraction Prompt**

You are an information extractor agent that operates in a workflow that solves a question step by step.
 You are given:
- The main question
- The sub-goal of the current step
- Part of the content of a webpage, you will be given the number of parts and the index of the current part
 **Your task:**
- Browse through the webpage content and look for the information that contains the answer to the sub-goal or the original question
- Extract that information **in its ORIGINAL FORM, don't paraphrase or modify the information**
- When you can't find the information from the current part, decide whether you should continue browsing the next part of the webpage
- The current date is **{current_date}**, be careful when the question requires updated information
 **Extra notes:**
- If no information founded, leave the information field as ""
- Make sure the answer can be clearly extracted from the information without any ambiguity
 — Main question:

{original_question} Sub-goal:

{sub_question} Webpage content:

{webpage_content_part}

---

**Parallel Browser.** The parallel browser block visits all URLs returned by the current search results concurrently. It uses the same prompts as General Browser with multi-threading.

**In-depth Browser.** The in-depth browser block selects up to three root URLs, extracts both relevant URLs and information from each page, maintains a queue of discovered pages, and continues visiting until the queue is empty or a predefined visit limit is reached.

## Root URL Selection Prompt

You are a helper agent that selects the root URLs to browse when solving a question step by step.
You are given:

- A question
- The sub-goal of the current step
- A list of URLs and their snippets
**Your tasks:**

- Select the root URLs that potentially have tabs and buttons to direct to pages with useful information relevant to the sub-goal
- If you think some URLs directly provide the information you need, you can also select them
- The current date is **{current_date}**, be careful when the question requires updated information **Extra notes:**

- **CRITICAL: Pay close attention to specific requirements in the question** (e.g., "official script", "official website", "primary source", "government data", etc.)
- **Prioritize URLs that match the specific source requirements mentioned in the question**
- If the question asks for "official" sources, prioritize URLs from official organizations, government sites, or primary sources over fan sites, transcripts, or secondary sources

{experiences} —
Question:

{question} Sub-goal:

{sub_goal} List of URLs and their snippets:

{list_of_urls_and_snippets}

## In-depth Browsing Prompt

You are a helper agent that browses the web to find useful information when solving a question step by step.
You are given:

- A question
- The sub-goal of the current step
- A part of the content of a webpage
**Your tasks:**

- Extract ALL information that could be relevant to answering the question or achieving the sub-goal
- Look for specific details like names, numbers, dates, relationships, lists, tables, and factual data
- Pay special attention to structured data (tables, lists, rosters, directories) that might contain answers
- Find links present in the webpage that can potentially direct to pages with useful information relevant to the sub-goal
- You are given the part index and the total number of parts of the webpage
- You need to decide whether to keep browsing the next part of the webpage if there is still part left
- Be generous in what you consider "relevant" - include information that might be indirectly useful
- The current date is **{current_date}**, be careful when the question requires updated information

—
Question:

{question}
Sub-goal:

> {sub_goal}
> Webpage content:
> {webpage_content}

## C.2 QUERY WRITER AND ORCHESTRATOR

The query writer is in charge of decomposing and generating the next query based on the current information summary and the original question.

---

**Next Query Writer Prompt**

You are an advisor agent that operates in a workflow that solves a question step by step using a search engine.
 You are given:

- A question
- A summary containing the current found information
 **Your tasks:**

- Review the current summary to see what information has already been found
- Identify what key information is still missing to answer the main question completely
- Write a comprehensive sub-goal that encompasses all the information agents can start to find given the summary's context
- If no summary provided, start with the first logical sub-goal needed to answer the main question, the first sub-goal can be exactly the same as the main question, if you think the main question is focused enough on one specific goal
 **Instructions:**

- Create a sub-goal that maximizes information gathering potential - don't limit the scope (Example: "Find a list ...")
- When there are several possible sub-goals, choose the one that is easier to reach using a search engine
- However, the new sub-goal still needs to be based on the current summary's context, missing context would mislead the workflow
- The sub-goal must encompass queries about all relevant information that can be discovered based on the current summary's context
- Stay focused on the main question - your sub-goal should be a necessary step toward answering it
- Use information from the current summary as context for the next sub-goal (e.g., if the summary identifies a city, use that city name in your next goal)
- The current date is **{current_date}**, be careful when the question requires updated information

**Key principle:** Always use specific information from the current summary in your next sub-goal rather than generic placeholders.
 —
 Question:
{question} Summary:
{current_summary}

---

The orchestrator is in charge of orchestrate the workflow from a pre-defined set of building blocks.

---

**Workflow Orchestration Prompt**

You are an orchestrator agent that designs search workflows to answer sub-questions using specialized building blocks.
 **Decision Logic:**

---

- Can sub-goal be answered without web searches? → **Thinker-Summarizer**
- Need new information from web? → **Searcher-Browser-Summarizer**

ENCOURAGE DIVERSE COMBINATIONS
 **Searcher Options**: fast_searcher, searcher, advanced_searcher

**Browser Options**: fast_browser, browser, advanced_browser, deep_browser
**Summarizer Options**: summarizer, advanced_summarizer
REQUIREMENTS
- **Function names must match exact 'block_name' from block list**
- **Output complete YAML** with "'yaml and "' tags
- **DO NOT modify the rest of the YAML template - only fill in the highlighted/placeholder parts**
- **ONLY use these 4 node types**: 'searcher', 'browser', 'summarizer', 'thinker' (and their variants)
- **DO NOT use**: 'verifier' or 'finalizer' - these are handled automatically
 **YAML template:**
"'yaml
{yaml_template} "' {experiences} **IMPORTANT**: Only modify the highlighted/placeholder sections in the template above. The rest of the YAML structure must remain unchanged.
 —
 **The original question:**
{original_question} **The current sub-goal:**
{question} **Current summary of the found information:**
{current_summary} **List of pre-defined building blocks:**
{list_of_building_blocks}

## C.3 YAML CONVERTER MODULE

In **FlowSeacher**, the orchestrator orchestrates the workflows by fill in a YAML template that already contains the verification logic as below:

### YAML Template

```
nodes:
**Your selected building blocks**
- name: sub_verifier
function: verifier
config:
verifier_variant: sub
- name: final_verifier
function: verifier
config:
verifier_variant: final
- name: finalizer
function: finalizer
 edges:

- START -  **The first building block to execute when solving the question**
**Your orchestrated edges**
- **The last building block to execute when solving the question** -  sub_verifier
- finalizer -  END
 conditional_edges:

- from: sub_verifier
condition: state.get('sub_verified')
```

```
routes:
- false: **The first building block to execute when solving the question**
- true: final_verifier
- from: final_verifier
condition: state.get('final_verified')
routes:
- false: END
- true: finalizer
```

## C.4 AGENTIC WORKFLOW CREATION AND EXECUTION

The workflow is created as a state graph with every agentic action altering part of fields of an overall state object.

**Agentic Workflow State**

**Overall State:**
- **messages:** list of messages (auto-append new messages)
- **current_sub_question:** string
- **current_sub_question_iteration:** integer
- **current_summary:** string
- **final_answer:** string
- **sub_verified:** boolean
- **final_verified:** boolean
- **searcher_state:** SearcherState (updated automatically)
- **browser_state:** BrowserState (updated automatically)
- **instruction_state:** InstructionState

**SearcherState:**
- **search_count:** integer
- **used_keywords:** list of strings
- **history_search_results:** list of SearchResult
- **search_results:** list of SearchResult
- **search_cache:** dictionary

**BrowserState:**
- **visit_count:** integer
- **visited_urls:** list of strings
- **history_found_references:** list of Reference
- **found_references:** list of Reference
- **visit_cache:** dictionary

**InstructionState:**
- **orchestrator_instructions:** list of strings
- **searcher_instructions:** list of strings
- **browser_instructions:** list of strings

When a workflow is successfully generated, a folder named "workspace" will be created with graph settings and executables. When a workflow is executed successfully, the final state is saved to the same "workspace" folder.

The structure of the folder is as follows:

---

**Workspace Structure after Successful Execution**

```
sample_workspace/
  after_state.json
  before_state.json
  graph.py
  graph.yaml
  run.py
```

---

### C.5 MEMORY STRUCTURE AND PROMPT INJECTION

**FlowSearcher** maintains a multi-level memory structured as the below schema:

---

**FlowSearcher Execution Memory**

**Overall State:**
- **execution_id**: string
- **question**: string
- **sub_question**: string
- **before_summary**: string
- **summary**: string
- **can_answer_sub_question**: boolean
- **can_answer_question**: boolean
- **workflow**: string (YAML representation)
- **searcher_execution_memory**: SearcherExecutionMemory (updated automatically)
- **browser_execution_memory**: BrowserExecutionMemory (updated automatically)
- **question_embedding**: list of floats
- **sub_question_embedding**: list of floats

**SearcherExecutionMemory:**
- **search_count**: integer
- **new_keywords_added**: list of strings
- **new_search_results**: list of SearchResult

**BrowserExecutionMemory:**
- **visit_count**: integer
- **new_visited_urls**: list of strings
- **new_references_found**: list of Reference

---

When a new sub-query is generated, we retrieve the relevant memory entries with a weighted sum of main question and sub-question similarity with default weights set to **0.5** and **0.5**.

Then we pass the graph-level and node-level traces to the instructor module. The trace schemas are as follows:

---

**Orchestrator Execution History Template**

```
Original question: {question}
Sub-goal: {sub_goal}
Before summary: {before_summary}
After summary: {after_summary}
Orchestrated workflow: {workflow}
Successful: {successful}
```

---

---

**Search Execution History Template**

```
Original question: {question}
Sub-goal: {sub_goal}
Search keywords: {search_keywords}
Search results: {search_results}
Successful: {successful}
```

---

**Browse Execution History Template**

```
Original question: {question}
Sub-goal: {sub_goal}
New found references: {new_found_references}
Successful: {successful}
```

Then, we pass the history traces of these format to the instructor module, gain the actionable experiences and inject them to corresponding prompts' placeholders.

## D CASE STUDY

In this section, we present two specific cases of our system performing different types of tasks with GPT-4o-mini backbone.

### D.1 EXAMPLE OF PERFORMING WEB NAVIGATION TASK

Below is an example execution of a web navigation task from GPT-4o-mini backbone. The workflow settings are First-hit Searcher + In-depth Browser + General Summarizer.

---

**Example Web Navigation Task**

**Original Question:** According to GitHub, when was the *Regression* label added to the oldest closed `numpy.polynomial` issue, in `MM/DD/YY` format?

**Summary from Previous Steps:** I found a filtered GitHub Issues view for the NumPy repository, showing issues labeled "06 – Regression" that are closed. This label is used to track regressions, i.e., cases where something that previously worked in NumPy became broken in a later version. The results are limited to the first page of closed regression issues, so only a subset is visible.

**Current Sub-goal:** Identify the oldest issue involving the component `numpy.polynomial`.

**Orchestrated Workflow:** First-hit Searcher + In-depth Browser + General Summarizer.

**Execution Process:**

- First-hit Searcher performs a search: `numpy polynomial issues Regression label GitHub`
- In-depth Browser processes the Regression-label issue page:
    - found page button: [20]
    - navigated to the target page
    - found link to the oldest issue: Issue #291
    - visited the link
    - extracted information
- General Summarizer incrementally updates the summary: ... The oldest closed Regression issue on GitHub is #291, opened on `Jun 5, 2012`. However, this issue does not appear to be directly relevant to `numpy.polynomial`. (A new iteration started.)

---

## D.2 Example of Performing Comprehensive Surveying Task

Below is an example execution of performing a comprehensive surveying task, the workflow settings are Advanced Searcher + Advanced Browser + Advanced Summarizer.

---

**Example Comprehensive Surveying Task**

**Original Question:** An African author tragically passed away in a tragic road accident. As a child, he'd wanted to be a police officer. He lectured at a private university from 2018 until his death. In 2018, this author spoke about writing stories that have no sell by date in an interview. One of his books was selected to be a compulsory school reading in an African country in 2017. Which years did this author work as a probation officer?

**Summary from Previous Steps:** The initial step is to deconstruct the user's query to identify the author in question. The key identifying facts are: an African author who died in a road accident, lectured at a private university from 2018 until his death, had a book become compulsory reading in an African country in 2017, and gave a specific interview in 2018. The final goal is to determine the years this author worked as a probation officer.

**Current Sub-goal:** Find a list of Arican writers who had a lecturer career.

**Orchestrated Workflow:** `Advanced Searcher` + `Advanced Browser` + `Advanced Summarizer`.

**Execution Process:**

- `Advanced Searcher` performs 25 searches: "African author lecturer died in accident", "Nigerian author book compulsory reading 2017", "Pius Adesanmi biography", "Pius Adesanmi career", "Pius Adesanmi probation officer employment history"

- `Advanced Browser` processes the 58 pages:
    - found informations from 25 pages:
    - Pius Adesanmi was a Nigerian-Canadian author and professor at Carleton University. He died in the Ethiopian Airlines Flight 302 crash on March 10, 2019. He was an active lecturer in 2018 until his death. His book "Naija No Dey Carry Last" was highly acclaimed and recommended for school curricula in Nigeria. A review of his CV and multiple biographies shows a career exclusively in academia and literature, with no mention of him working as a probation officer.

- `Advanced Summarizer` ensembles and updates the summary: The author described is identified as Pius Adesanmi based on his profile as a Nigerian author, lecturer, the timing of his death in an accident, and the status of his book. After a thorough review of his employment history, there is no evidence that he ever worked as a probation officer. The premise of the original question appears to be incorrect. (A new iteration started.)

---

# E  Acknowledgment on LLM Usage

We acknowledged the use of large language models (LLMs) specifically for language polishing, rephrasing, and improving readability, which helped us present our demonstrations more clearly. These tools were employed solely for stylistic refinement and did not contribute to the conceptualization, design, or methodological development of this work. All ideas, experiments, analyses, and conclusions remain entirely the result of the authors' own efforts.

